# Online Prediction of Ship Coupled Heave-Pitch Motions in Irregular Waves Based on a Coarse-and-Fine Tuning Fixed-Grid Wavelet Network

Baigang Huang [1,2], Jianjun Jiang [1] and Zaojian Zou [2,3,*]

1   Science and Technology on Communication Information Security Control Laboratory,
    No. 36 Research Institute of CETC, Jiaxing 314033, China; ballic@alumni.sjtu.edu.cn (B.H.);
    manluk@zju.edu.cn (J.J.)
2   School of Naval Architecture, Ocean and Civil Engineering, Shanghai Jiao Tong University,
    Shanghai 200240, China
3   State Key Laboratory of Ocean Engineering, Shanghai Jiao Tong University, Shanghai 200240, China
*   Correspondence: zjzou@sjtu.edu.cn

**Abstract:** A method based on a coarse- and fine-tuning fixed-grid wavelet networks is presented for online prediction of the coupled heave-pitch motions of a ship in irregular waves. The online modeling method contains two processes, i.e., coarse tuning and fine tuning. The coarse tuning is used to select the important wavelet terms, while the fine tuning is only used to compute the related coefficients of the selected wavelet terms. The Givens transformation algorithm is applied to realize the fine-tuning process. Due to the continuous fine-tuning process, the computational efficiency is improved significantly. Both simulation data and experimental data are used to verify the modeling method. The prediction results illustrate that the method has the ability to online predict the coupled heave-pitch motions of a ship in irregular waves.

**Keywords:** ship heave-pitch motions; irregular waves; online prediction; coarse-and-fine tuning fixed grid wavelet network

## 1. Introduction

Due to the environmental disturbances such as wind, waves and current, it is very difficult to establish a precise mathematical model for ship motion at sea [1]. Over many years, a variety of methods have been proposed and used to predict ship motion. The classic method is Kalman filtering technique; but to use it, the state equation of ship motion must be known first [2,3]. Another commonly used method is time-series-analysis techniques, such as autoregression (AR) model and autoregressive moving average (ARMA) model, which are linear analysis methods. In many cases, they are not suitable for modeling nonlinear ship motion. The computational fluid dynamics (CFD) method is another important method for modeling and predicting ship motion at sea, but ship geometry and environmental information are prerequisites for using it, and it is extremely time consuming.

In recent years, with the development of artificial intelligence (AI) technology, the related machine-learning technology has been widely used in the naval architecture and shipping industry. Back-propagation neural networks have been used to predict ship motion [4,5]. The radial-basis-function neural network was proposed to predict vessels' heave motion [6], a support vector machine was proposed to predict ship motion [7–9], and the adaptive wavelet network was used to predict short term ship motion [10,11]. With the development of deep learning, it has also been applied in ship-motion prediction; a hybrid deep-learning and ARIMA model was used for ships' roll motion prediction [12]. Recurrent neural network (RNN) is used for predicting the vertical acceleration of a large-scale ship [13]. Convolutional neural networks (CNN) and long short-term memory (LSTM) neural networks are also used for ship-motion prediction. The dual-pass LSTM network is

used for ship-heave prediction by Hu et al. [14]. CNN with LSTM is used for roll-motion prediction of unmanned surface vehicles (USVs) by Zhang et al. [15]. Deep-neural-network models usually contain many training parameters; thus, they often need a large number of training samples. Neural networks can theoretically fit arbitrary nonlinear functions with arbitrary precision [16]; because of this advantage, they are also used in ship motion modeling and parameter identification. Haddara and Xu used the random-decrement technique and neural networks to identity the parameters of coupled heave-pitch motion equations [17]. Yin et al. used neural networks to predict ship's roll motion during maneuvering [18]. Hou et al. used support vector machines to identify the parameters of coupled heave-pitch motions of a ship in regular waves [19]. However, many neural networks trained by gradient-type algorithms are liable to fall into local optimum, and the number of neurons in the hidden layers is not easy to predetermine, especially for online modeling, as the structure of neural networks is difficult to change online.

Among the six degrees of freedom ship motions, the coupled heave-pitch motions are most important for ships' seakeeping performance, since they are directly related to green water on deck, slamming, propeller racing, etc. Besides, the prediction of coupled heave-pitch motion is very important for motion compensation in marine operations, such as the landing of air vehicles on ship-based platforms, the launching of shipboard weapons, and lifting operations at sea; it is also important for ships' motion control, especially by using model-predictive control algorithms. Therefore, the prediction of coupled heave-pitch motions of ships in waves has attracted great attention from the ship hydrodynamics community, and many studies have been carried out in this field. However, there are still some challenging issues, such as the online prediction of coupled heave-pitch motions of a ship in waves, which is worth studying.

In this paper, a coarse-and-fine tuning fixed-grid wavelet network (CFT-FGWN) is used to establish the prediction model of coupled heave-pitch motions of a ship in irregular waves, aiming at online prediction of the coupled heave-pitch motions of a ship sailing at sea. The CFT-FGWN was first presented in Huang et al. [20,21] for the online prediction of ships' roll motion. The modeling method of CFT-FGWN can online change the structure of the wavelet network and avoid local optima, and usually converges fast. This modeling method was used for predicting ships' roll motion in regular waves [20] and in irregular waves [21]; there was only once fine tuning in these studies. In this paper, continuous fine-tuning is implemented for two degrees of freedom ship motions, i.e., heave and pitch, in irregular waves, which makes the computing efficiency greatly improved. The results show that the online modeling method can effectively predict the coupled heave-pitch motions of a ship in irregular waves.

## 2. Fixed-Grid Wavelet Networks for a Multiple-Inputs Multiple-Outputs System

In general, a multiple-inputs multiple-outputs (MIMO) system can be represented as

$$
\begin{cases}
y_1(t) = f_1(x_1(t), x_2(t), \cdots, x_n(t)) + e_1(t) \\
y_2(t) = f_2(x_1(t), x_2(t), \cdots, x_n(t)) + e_2(t) \\
\quad\quad\quad\quad\quad \vdots \\
y_m(t) = f_m(x_1(t), x_2(t), \cdots, x_n(t)) + e_m(t)
\end{cases}
\tag{1}
$$

where $\boldsymbol{x}(t) = [x_1(t), x_2(t), \cdots, x_n(t)]^{\mathrm{T}}$ and $\boldsymbol{y}(t) = [y_1(t), y_2(t), \cdots, y_m(t)]^{\mathrm{T}}$ are the system input vector and system output vector, respectively; $\boldsymbol{e}(t) = [e_1(t), e_2(t), \cdots, e_m(t)]^{\mathrm{T}}$ is the system noise vector; $n$ and $m$ are the numbers of input variables and output variables of the system, respectively. A neural network is a typical MIMO system model. By simple mathematical decomposition, it can also be expressed in the form of Equation (1). In fact, each MIMO system can be thought of as a combination of multiple-inputs single-output (MISO) systems, and each MISO system can be described by a multivariate nonlinear function. In this paper, a combination of wavelet-basis functions is used to represent this nonlinear function.

According to wavelet theory, the multivariate nonlinear function $f(x_1(t), x_2(t), \cdots, x_n(t))$ can be expressed as [20–24]

$$
\begin{aligned}
f(x_1(t), x_2(t), \cdots, x_n(t)) = c_0 &+ \sum_{i=1}^{n} c_i \psi^{[1]}(x_i(t)) + \sum_{1<i<j\leq n} c_{i,j} \psi^{[2]}(x_i(t), x_j(t)) + \cdots \\
&+ \sum_{1<i_1<\cdots<i_m\leq n} c_{i,i_2,\cdots,i_m} \psi^{[m]}(x_{i_1}(t), x_{i_2}(t), \cdots x_{i_m}(t)) + \cdots + e(t) \quad i,j \in Z
\end{aligned} \tag{2}
$$

where $c_0$ is a constant component representing the intrinsic varying trend; $\psi^{[1]}(\cdot)$, $\psi^{[2]}(\cdot)$ and $\psi^{[m]}(\cdot)$ are the univariate, bivariate and $m$-dimensional wavelet-basis functions, respectively; and $c_i$, $c_{i,j}$ and $c_{i_1,i_2,\cdots,i_m}$ are the coefficients of the corresponding wavelet-basis functions. Equation (2) can be seen as the structure of the fixed grid wavelet network. Due to this variety of wavelet-basis functions, different wavelet-basis functions can be selected for different nonlinear systems. Especially for strongly nonlinear systems, the fixed-grid wavelet network usually converges very quickly.

In order to predict the coupled heave-pitch motions of a ship with two degrees of freedom, the model for this multivariate nonlinear system can be expressed as

$$
\begin{cases}
f_1(x(t)) = c_0 + \sum_{i=1}^{n} c_i \psi^{[1]}(x_i(t)) + \sum_{1<i<j\leq n} c_{i,j} \psi^{[2]}(x_i(t), x_j(t)) + \cdots \\
\quad + \sum_{1<i_1<\cdots<i_m\leq n} c_{i_1,i_2,\cdots,i_m} \psi^{[m]}(x_{i_1}(t), x_{i_2}(t), \cdots x_{i_m}(t)) + \cdots + e_1(t) \\
f_2(x(t)) = c'_0 + \sum_{i=1}^{n} c'_i \psi'^{[1]}(x_i(t)) + \sum_{1<i<j\leq n} c'_{i,j} \psi'^{[2]}(x_i(t), x_j(t)) + \cdots \\
\quad + \sum_{1<i_1<\cdots<i_m\leq n} c'_{i_1,i_2,\cdots,i_m} \psi'^{[m]}(x_{i_1}(t), x_{i_2}(t), \cdots x_{i_m}(t)) + \cdots + e_2(t)
\end{cases} \tag{3}
$$

where $f_1(\cdot)$ and $f_2(\cdot)$ are the nonlinear functions of heave motion and pitch motion, respectively. Thus, the key problem of the modeling method is how to determine these wavelet terms and their corresponding coefficients. The coarse tuning has the ability to select more important univariate wavelet-basis functions, bivariate wavelet-basis functions and other wavelet-basis functions, while the fine tuning is used to determine the coefficients of the corresponding terms.

## 3. Coarse-Tuning and Fine-Tuning Algorithm

The coarse-tuning process can choose different important terms in the wavelet library, while the fine tuning only computes the coefficients of the selected important terms. The flow chart of the coarse-tuning and fine-tuning algorithm for the coupled heave-pitch motion modeling is shown in Figure 1. The dataset of coupled heave-pitch motions is a time series. Before modeling, this dataset must be normalized. The sliding data window is a data sequence; the sampling data entering the queue first go out first. The data in the sliding data window are used to establish the prediction model of the coupled heave-pitch motions. The wavelet library contains many different types of wavelet-basis functions; the variety is rich; the representation ability is strong. The orthogonal least squares (OLS) algorithm and error reduction ratio (ERR) criterion can quickly find the more important wavelet-basis functions in the library. First the most important, then the second most important, one after the other until the model meets the accuracy requirement of the design. After finding out the important wavelet-basis functions, the model is fine-tuned by the Givens transformation algorithm, that is, the corresponding coefficients of the selected wavelet-basis functions are calculated. The combination of coarse tuning and fine tuning can determine the corresponding nonlinear model effectively. With the sliding data window moving, the prediction model of coupled heave-pitch motions can be established online.

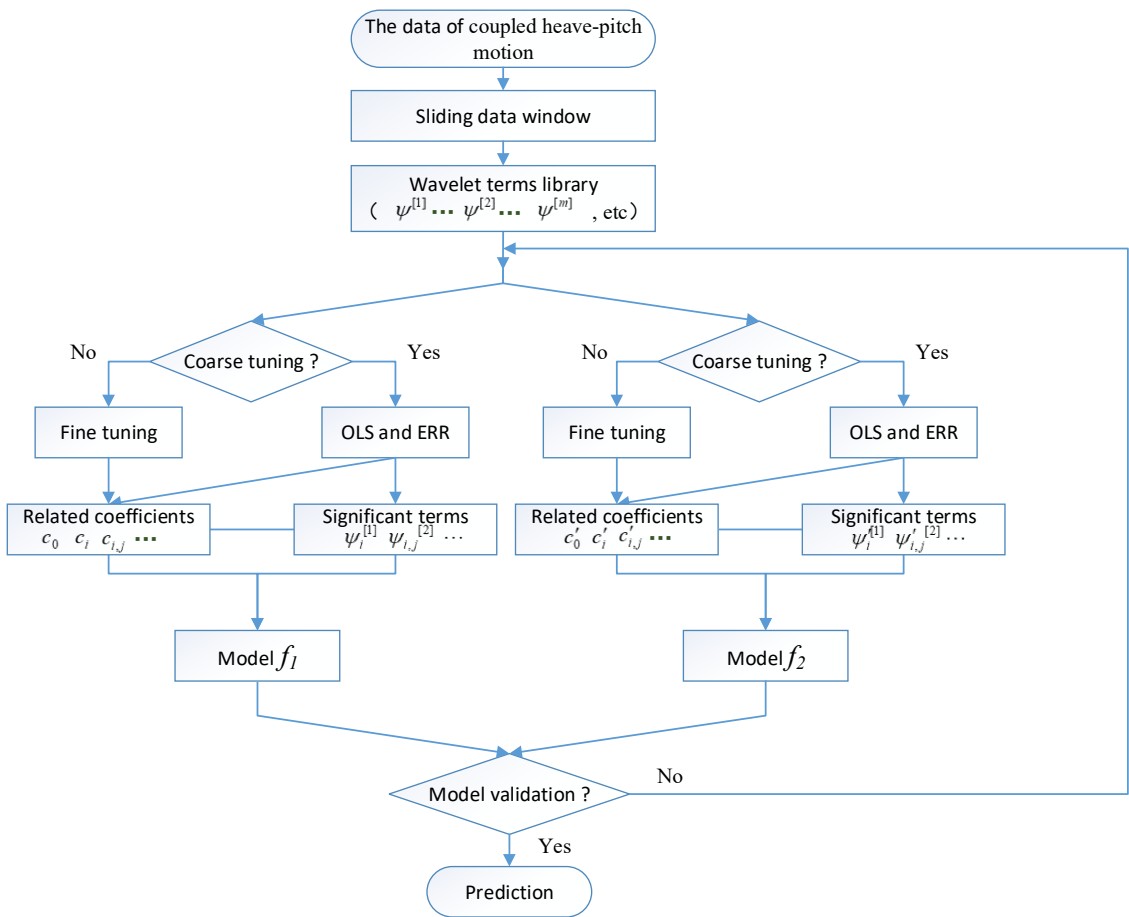

**Figure 1.** Flow chart of the coarse-tuning and fine-tuning algorithm for the coupled heave-pitch motions.

The OLS algorithm and ERR criterion are applied to realize the coarse-tuning process. The $\text{ERR}_i$ caused by $q_i(t)$ is defined as [20,21,25,26]

$$\text{ERR}_i = \frac{\langle Y(t), q_i(t)\rangle^2}{\langle Y(t), Y(t)\rangle \langle q_i(t), q_i(t)\rangle} \times 100\%, \qquad i = 1, 2, \cdots, \tag{4}$$

where $Y(t)$ is the output vector; $q_i(t)$ is the orthogonal column of the corresponding wavelet terms. The greater the $\text{ERR}_i$, the more important the corresponding wavelet term. According to the $\text{ERR}_i$ and model validation conditions, the structure of the fixed-grid wavelet networks can be easily determined.

The Givens transformation algorithm is applied to realize the fine-tuning process. The task of fine tuning is to online compute the coefficients of the selected wavelet terms, and the recursive transformation can be expressed as follows [20,21]:

$$\begin{bmatrix} R_{m(t)\times m(t)}(t) & V_{m(t)\times 1}(t) \\ \psi_1(t+1), \cdots, \psi_{m(t)}(t+1) & y(t+1) \end{bmatrix} \xrightarrow[\text{rotation}]{\text{Givens}} \begin{bmatrix} R_{m(t)\times m(t)}(t+1) & V_{m(t)\times 1}(t+1) \\ O_1 & \xi(t+1) \end{bmatrix}$$
$$\downarrow$$
$$\begin{bmatrix} R_{m(t)\times m(t)}(t+1) & V_{m(t)\times 1}(t+1) \\ O_1 & O_2 \end{bmatrix} \tag{5}$$

where $m(t)$ is the number of the significant components at the instant $t$ after the coarse tuning; $R_{m(t)\times m(t)}(t)$ is a unit upper triangular matrix at the instant $t$; $V_{m(t)\times 1}(t)$ is a column vector at the instant $t$ that depends on the system output; $[\psi_1(t+1), \cdots, \psi_{m(t)}(t+1)]$ are the outputs of wavelet terms depending on the system input at the instant $t+1$; $y(t+1)$

is the system output at the instant $t + 1$; $O_1$ and $O_2$ are the corresponding zero matrixes, respectively; $\xi(t + 1)$ represents the error at the instant $t + 1$. The estimation of $\Theta_{m(t) \times m(t)}(t)$ is equal to $R_{m(t) \times m(t)}^{-1}(t) V_{m(t) \times 1}(t)$.

For the sparse matrix on the left-hand side of Equation (5), the Givens transformation algorithm is described as follows:

$$
\begin{array}{c}
\text{For } i = 1 : m(t) \\
r = \sqrt{A^2(i,i) + A^2(m(t) + 1, i)}; \\
c = \frac{A(i,i)}{r} , \ s = \frac{A(m(t)+1,i)}{r} ; \\
\text{For } j = i : m(t) + 1 \\
A(i,j) = cA(i,j) + sA(m(t) + 1, j) ; \\
A(m(t) + 1, j) = -sA(i,j) + cA(m(t) + 1, j) ; \\
\text{End} \\
\text{End}
\end{array}
\tag{6}
$$

where $\mathbf{A}(i,j)$ represents the $i$-th row and $j$-th column element of the sparse matrix. Yet there is a drawback: if the sparse matrix is decomposed continuously by the Givens transformations, the first element of matrix $R(t)$ will tend to infinite, which may cause the ill-condition of matrix $R(t)$. This problem can be overcome by the back-operation, which is a reverse operation to eliminate the impact of the oldest data in the sliding data window [27,28]. The algorithm is as follows:

$$
\begin{array}{c}
\text{For } i = 1 : m(t) \\
r = \mathbf{A}(i,i) ; \\
c = \frac{\sqrt{\mathbf{A}^2(i,i) + \mathbf{A}^2(m(t)+1,i)}}{r} , \ s = \frac{\mathbf{A}(m(t)+1,i)}{r} ; \\
\text{For } j = i : m(t) + 1 \\
\mathbf{A}(i,j) = \frac{\mathbf{A}(i,j) - s\mathbf{A}(m(t)+1,j)}{c} ; \\
\mathbf{A}(m(t) + 1, j) = c\mathbf{A}(m(t) + 1, j) - s\mathbf{A}(i,j) ; \\
\text{End} \\
\text{End}
\end{array}
\tag{7}
$$

By eliminating the impact of the oldest data and adding the impact of the latest data, the parameter vector can be updated in an iterative way. In order to ensure the flexibility of the model's structure, the maximum number for the continuous fine-tuning should not be set too large.

## 4. Simulation Results and Discussion

Based on the simulation data and the experimental data of coupled heave-pitch motions in irregular waves, the proposed CFT-FGWN is used to establish the forecast model of coupled heave-pitch motions. All the programs are executed by MATLAB R2016b under the condition of Intel® Core™ i5-3230 (CPU) and 8.00 GB memory (RAM).

### 4.1. Prediction Results Based on Simulation Data

The coupled heave-pitch motions of a ship in irregular waves can be represented by two coupled second-order linear ordinary differential equations, which can be written in the following form [29]:

$$
\begin{cases}
(m_0 + m_{33})\ddot{z} + N_{33}\dot{z} + C_{33}z + (m_{35} - m_0 x_G)\ddot{\theta} + N_{35}\dot{\theta} + C_{35}\theta = \tau_3 \\
(I_{22} + m_{55})\ddot{\theta} + N_{55}\dot{\theta} + C_{55}\theta + (m_{53} - m_0 x_G)\ddot{z} + N_{53}\dot{z} + C_{53}z = \tau_5
\end{cases}
\tag{8}
$$

where $z$ and $\theta$ are the heave displacement and pitch angle, respectively; $x_G$ is the longitudinal coordinate of the center of gravity of the ship; $m_0$ is the mass; $I_{22}$ is the pitch moment of inertia; $m_{33}$ and $m_{55}$ are the added mass and added moment of inertia, respectively; $N_{33}$ and $N_{55}$ are the damping coefficients, respectively; $C_{33}$ and $C_{55}$ are the restoring force

coefficients, respectively; $m_{ij}$, $N_{ij}$, and $C_{ij}$ ($i, j = 3, 5$) are the coupled coefficients; $\tau_3$ and $\tau_5$ are the wave exciting force and moment, respectively.

After the normalization of the second-order derivative terms in Equation (8), the normalized differential equations of coupled heave-pitch motions can be obtained [19]:

$$
\begin{cases}
\ddot{z} + b_{33}\dot{z} + b_{35}\dot{\theta} + r_{33}z + r_{35}\theta = \tau'_3 \\
\ddot{\theta} + b_{53}\dot{z} + b_{55}\dot{\theta} + r_{53}z + r_{55}\theta = \tau'_5
\end{cases}
\tag{9}
$$

where

$$
\begin{pmatrix} b_{33} & b_{35} \\ b_{53} & b_{55} \end{pmatrix} = \begin{pmatrix} m_0 + m_{33} & m_{35} - m_0 x_G \\ m_{53} - m_0 x_G & I_{22} + m_{55} \end{pmatrix}^{-1} \begin{pmatrix} N_{33} & N_{35} \\ N_{53} & N_{55} \end{pmatrix}
$$

$$
\begin{pmatrix} r_{33} & r_{35} \\ r_{53} & r_{55} \end{pmatrix} = \begin{pmatrix} m_0 + m_{33} & m_{35} - m_0 x_G \\ m_{53} - m_0 x_G & I_{22} + m_{55} \end{pmatrix}^{-1} \begin{pmatrix} C_{33} & C_{35} \\ C_{53} & C_{55} \end{pmatrix}
$$

$$
\begin{pmatrix} \tau'_3 \\ \tau'_5 \end{pmatrix} = \begin{pmatrix} m_0 + m_{33} & m_{35} - m_0 x_G \\ m_{53} - m_0 x_G & I_{22} + m_{55} \end{pmatrix}^{-1} \begin{pmatrix} \tau_3 \\ \tau_5 \end{pmatrix}
$$

In order to simulate the coupled heave-pitch motions in irregular waves, the JON-SWAP (Joint North Sea Wave Project) wave spectrum is used to generate the irregular waves. It is defined by [30]

$$
S(f) = \frac{5H_s^2 f_m^4}{16 f^5 \gamma^{\frac{1}{3}}} \exp\left(-\frac{5f_m^4}{4f^4}\right) \gamma^{\exp\left[-\frac{(f-f_m)^2}{2\sigma^2 f_m^2}\right]}
\tag{10}
$$

where $f$ is the wave frequency, $H_s$ is the significant wave height, $f_m$ is the peak frequency, $\gamma$ is the peak enhancement factor, and $\sigma$ is the shape parameter. The values of the parameters listed in Table 1 are chosen for simulation of the irregular waves [30]. The wave frequency range is between 0.01 Hz and 2 Hz, and the sampling interval is 0.01 Hz. There are 200 frequency components, and the simulated irregular waves are shown in Figure 2.

**Table 1.** The parameters of JONSWAP wave spectrum.

| Parameter | $H_s$ | $f_m$ | $\gamma$ | $\sigma$ |
|---|---|---|---|---|
| Value | 5 cm | 0.7 Hz | 3.3 | 0.07, $f < f_m$ <br> 0.09, $f \geq f_m$ |

The ship model is the R-Class Icebreaker model, and the principal particulars of the model are listed in Table 2. According to the strip theory, the related coefficients of Equation (9) are obtained and given in Table 3. More details can be found in Xu [30].

**Table 2.** Principal particulars of the R-Class Icebreaker model.

| Parameter | Value |
|---|---|
| Length between perpendiculars | 2.1985 m |
| Length of waterline | 2.3250 m |
| Waterline beam at midships | 0.4840 m |
| Draft at midships | 0.1735 m |
| Displacement | 121.6 kg |

**Table 3.** The parameters in Equation (9).

| Parameter | $b_{33}$ | $b_{35}$ | $r_{33}$ | $r_{35}$ | $b_{53}$ | $b_{55}$ | $r_{53}$ | $r_{55}$ |
|---|---|---|---|---|---|---|---|---|
| Value | 2.8235 | 0.1565 | 34.0961 | 0.2229 | 0.5797 | 2.6343 | 0.6261 | 30.8090 |

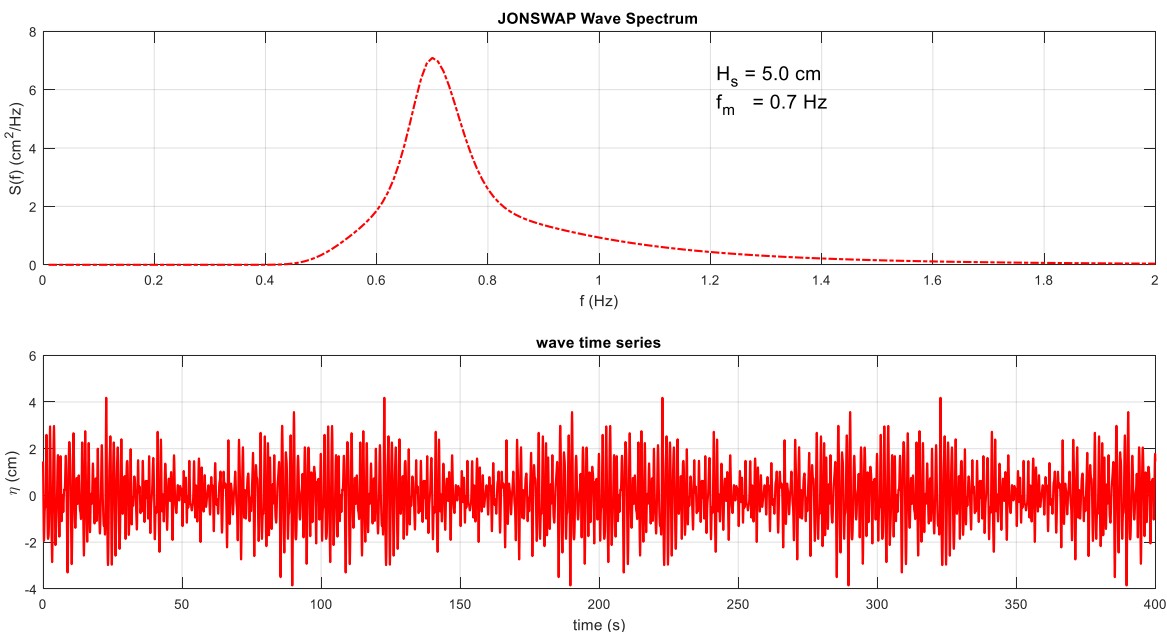

**Figure 2.** The JONSWAP wave spectrum and related irregular waves.

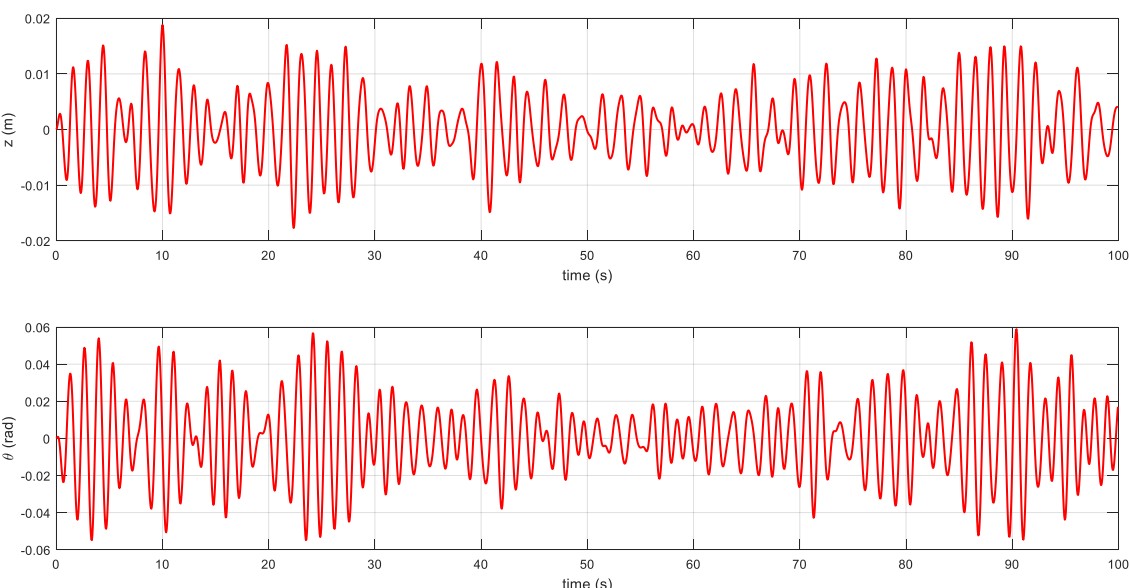

**Figure 3.** The coupled heave-pitch motions of the R-Class Icebreaker model in irregular waves.

Equation (9) is discretized, and it gives:

$$\begin{cases} z(k) = f_1(z(k-1), z(k-2), \tau'_3(k-1), \theta(k-1), \theta(k-2), \tau'_5(k-1)) \\ \theta(k) = f_2(z(k-1), z(k-2), \tau'_3(k-1), \theta(k-1), \theta(k-2), \tau'_5(k-1)) \end{cases} \quad (11)$$

where $k$ denotes the $k$-th sampling instant.

It is easy to know that the six input variables in Equation (11) may be the most important. According to the prediction results, $\{z(k-1), z(k-2), \theta(k-1), \theta(k-2)\}$ are the most significant terms, and the nonlinear autoregressive (NAR) model is enough to

describe the essential characteristics of the coupled heave-pitch motions in irregular waves. The structure of the fixed-grid wavelet networks is set as follows:

$$y_m(k) = f_m(z(k-1), z(k-2), \theta(k-1), \theta(k-2))$$
$$= \sum_{i=1}^{4} f_i(x_i(k)) + \sum_{1 \leq i < j \leq 4} f_{ij}(x_i(k), x_j(k)) + e_m(k) \tag{12}$$

where

$$y_1(k) = z(k), \ y_2(k) = \theta(k); \ x_i(k) = z(k-i) \ (i = 1,2), \ x_i(k) = \theta(k-i) \ (i = 3,4).$$

The functional components $f_i$ and $f_{ij}$ in Equation (12) can be fitted by the one- and two-dimensional Mexican-hat radial wavelets. The maximum number of the continuous fine-tuning is not larger than 5. The coarsest resolutions $j_1$ and the finest resolutions $j_{\max1}$ are set to 0 and 3 in the one-dimensional radial wavelets, and the coarsest resolutions $j_2$ and the finest resolutions $j_{\max2}$ are set to 0 and 2 in the two-dimensional radial wavelets. The sliding data window contains the data of coupled heave-pitch motions for training the wavelet model. The coarse training condition for the wavelet model of heave motion is that the other significant terms must be added in the model when $\sum_i ERR_i < 0.99$ or $RMSE_{training} > 0.012$, while the coarse training condition for the wavelet model of pitch motion is $\sum_i ERR_i < 0.99$ or $RMSE_{training} > 0.008$, where $RMSE_{training}$ represents the root mean square error (*RMSE*) of the training data. These values are determined by the trial-and-error approach, and the *RMSE* is defined as

$$RMSE = \sqrt{\frac{\sum_{i=1}^{N} (y_i - \hat{y}_i)^2}{N}} \tag{13}$$

where $N$ represents the number of samples; $y_i$ is the sample data, and $\hat{y}_i$ is the estimated data by the model.

The identified model is given as

$$
\begin{cases}
z(k) = c_1 \cdot \psi_{0;-1,4}^{[2]}(z(k-2), z(k-1)) + c_2 \cdot \psi_{1;-2,-2}^{[2]}(z(k-1), \theta(k-1)) \\
\quad + c_3 \cdot \psi_{2;-1,4}^{[2]}(z(k-2), z(k-1)) + c_4 \cdot \psi_{0;1,-1}^{[2]}(z(k-2), z(k-1)) \\
\quad + c_5 \cdot \psi_{1;2}(z(k-2)) + c_6 \cdot \psi_{2;6,2}^{[2]}(z(k-1), \theta(k-1)) + c_7 \cdot \psi_{3;0}(z(k-1)) \\
\theta(k) = d_1 \cdot \psi_{0;-1,4}^{[2]}(\theta(k-2), \theta(k-1)) + d_2 \cdot \psi_{2;4,-3}^{[2]}(\theta(k-2), \theta(k-1)) \\
\quad + d_3 \cdot \psi_{2;-1,3}^{[2]}(\theta(k-2), \theta(k-1)) + d_4 \cdot \psi_{1;5,-2}^{[2]}(z(k-2), z(k-1)) \\
\quad + d_5 \cdot \psi_{0;1,-1}^{[2]}(\theta(k-2), \theta(k-1)) + d_6 \cdot \psi_{2;7,-1}^{[2]}(\theta(k-2), \theta(k-1)) \\
\quad + d_7 \cdot \psi_{0;-3,4}^{[2]}(z(k-2), z(k-1)) + d_8 \cdot \psi_{1;5,-1}^{[2]}(\theta(k-2), \theta(k-1))
\end{cases} \tag{14}
$$

where the related coefficients are given in Table 4. The outputs of the identified model are inversely normalized to obtain the original system outputs.

**Table 4.** The coefficients in Equation (14).

| Coefficient | $c_1$ | $c_2$ | $c_3$ | $c_4$ | $c_5$ | $c_6$ | $c_7$ | |
|---|---|---|---|---|---|---|---|---|
| **Value** | $-34.1789$ | 0.3242 | $-0.1250$ | $-1.2824$ | $-0.0652$ | $-0.0677$ | $-0.0100$ | |
| **Coefficient** | $d_1$ | $d_2$ | $d_3$ | $d_4$ | $d_5$ | $d_6$ | $d_7$ | $d_8$ |
| **Value** | $-35.6518$ | 4.3020 | $-0.1215$ | 60.1121 | $-0.9676$ | 81.3790 | $-386.9376$ | 12.1714 |

The results of initial one-step ahead prediction are depicted in Figures 4 and 5, where the solid line shows the training part, and the dash line shows the testing part. The RMSE of the identified wavelet model is given in Table 5.

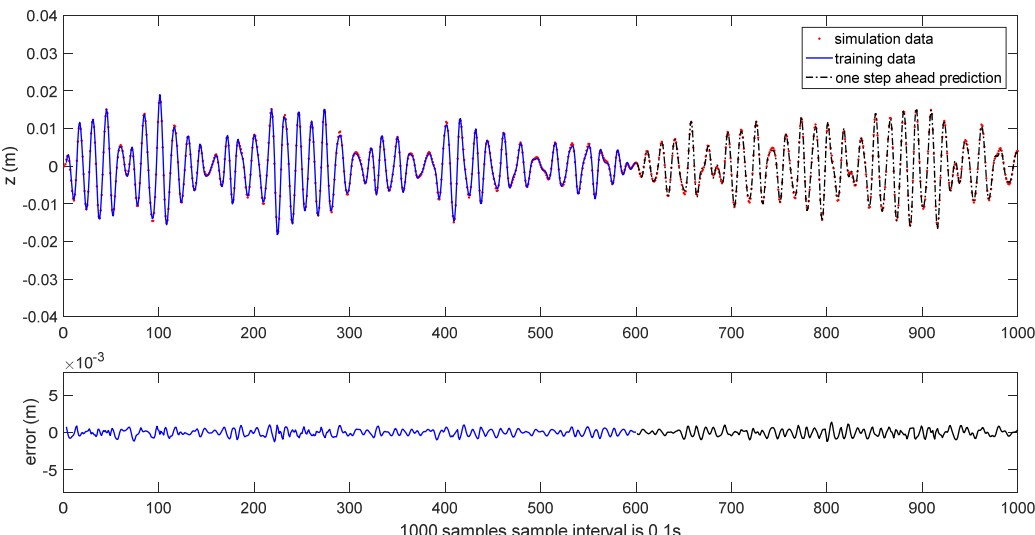

**Figure 4.** The one-step-ahead prediction of heave motion in irregular waves for the R-Class Icebreaker model.

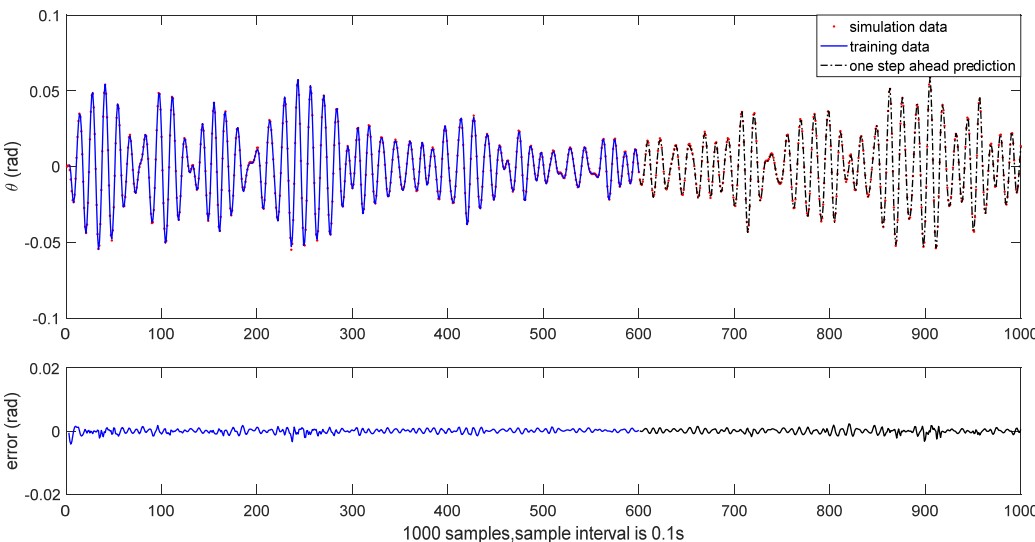

**Figure 5.** The one-step-ahead prediction of pitch motion in irregular waves for the R-Class Icebreaker model.

**Table 5.** The RMSE of the identified wavelet model.

| **Motion** | $RMSE_{training}$ | $RMSE_{testing}$ |
|---|---|---|
| Heave (m) | $4.1442 \times 10^{-4}$ | $4.7638 \times 10^{-4}$ |
| Pitch (rad) | $7.1254 \times 10^{-4}$ | $7.3482 \times 10^{-4}$ |

The multi-steps-ahead prediction can be expressed as

$$\hat{y}(k+s) = f(\hat{z}(k+s-1), \hat{z}(k+s-2), \hat{\theta}(k+s-1), \hat{\theta}(k+s-2)) \tag{15}$$

where $s$ represents the prediction instant.

The 20-steps-ahead prediction results are depicted in Figures 6 and 7. As it can be seen in these figures, the prediction performance is acceptable. Due to the accumulation of data error, the prediction performance is degraded.

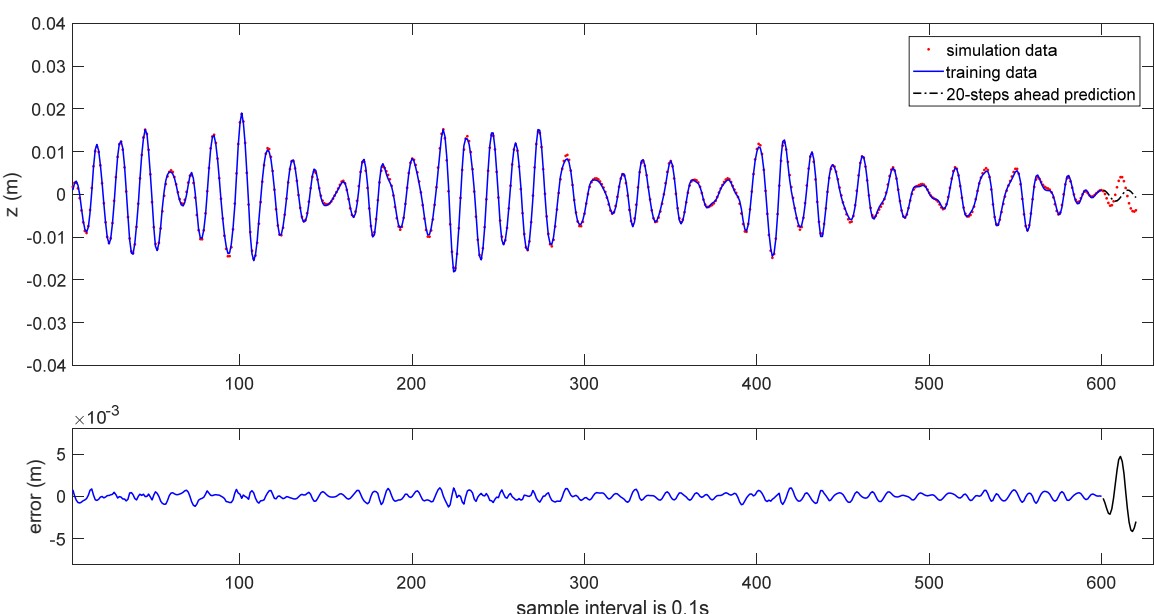

**Figure 6.** The 20-steps-ahead prediction of heave motion in irregular waves for the R-Class Icebreaker model.

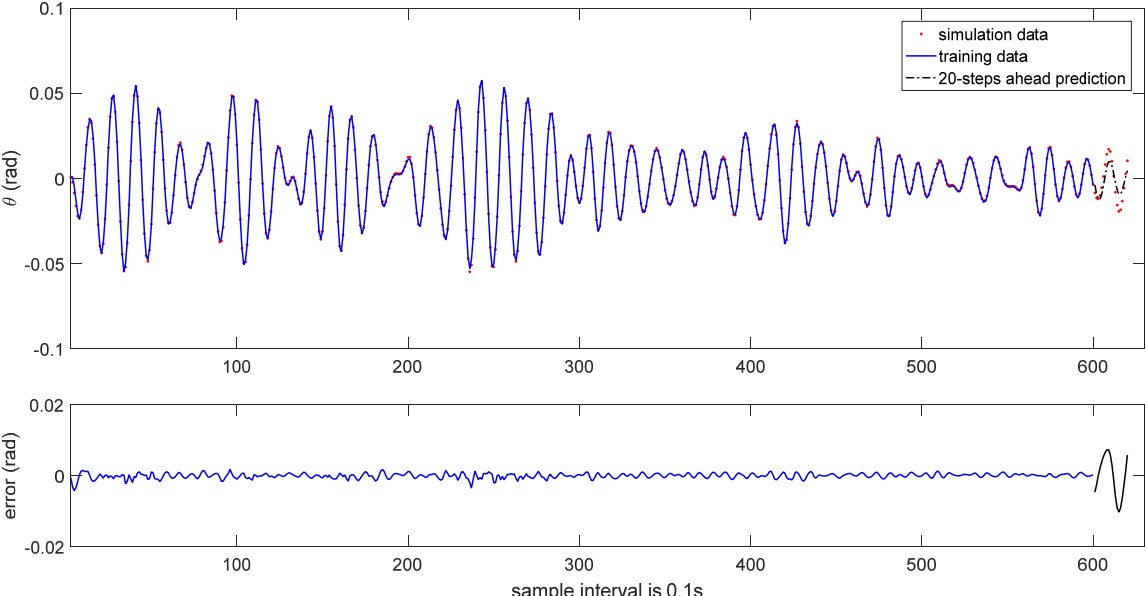

**Figure 7.** The 20-steps-ahead prediction of pitch motion in irregular waves for the R-Class Icebreaker model.

When the sliding data window receives new sample data, the above modeling process will be repeated. For example, as the sliding data window moves to the sampling instant $k$ = 687, the corresponding identified model is given as follows:

$$
\begin{cases}
z(k) = c_1 \cdot \psi_{0;-1,4}^{[2]}(z(k-2), z(k-1)) + c_2 \cdot \psi_{1;-2,-2}^{[2]}(z(k-1), \theta(k-1)) \\
\quad + c_3 \cdot \psi_{2;-1,4}^{[2]}(z(k-2), z(k-1)) + c_4 \cdot \psi_{0;1,-1}^{[2]}(z(k-2), z(k-1)) \\
\quad\quad + c_5 \cdot \psi_{1;2}(z(k-2)) + c_6 \cdot \psi_{2;6,2}^{[2]}(z(k-1), \theta(k-1)) \\
\theta(k) = d_1 \cdot \psi_{0;-1,4}^{[2]}(\theta(k-2), \theta(k-1)) + d_2 \cdot \psi_{0;-1,1}^{[2]}(\theta(k-2), \theta(k-1)) \\
\quad + d_3 \cdot \psi_{0;1,1}^{[2]}(\theta(k-2), \theta(k-1)) + d_4 \cdot \psi_{1;0,2}^{[2]}(z(k-2), z(k-1)) \\
\quad + d_5 \cdot \psi_{2;5,1}^{[2]}(\theta(k-2), \theta(k-1)) + d_6 \cdot \psi_{1;-3,4}^{[2]}(z(k-2), z(k-1)) \\
\quad + d_7 \cdot \psi_{2;-3,5}^{[2]}(\theta(k-2), \theta(k-1)) + d_8 \cdot \psi_{2;-2,5}^{[2]}(z(k-2), z(k-1))
\end{cases}
\tag{16}
$$

where the related coefficients are listed in Table 6.

**Table 6.** The coefficients in Equation (16).

| Coefficient | $c_1$ | $c_2$ | $c_3$ | $c_4$ | $c_5$ | | $c_6$ | |
|---|---|---|---|---|---|---|---|---|
| **Value** | $-36.0593$ | $0.4879$ | $-0.1297$ | $-1.1814$ | $-0.0633$ | | $-0.0577$ | |
| **Coefficient** | $d_1$ | $d_2$ | $d_3$ | $d_4$ | $d_5$ | $d_6$ | $d_7$ | $d_8$ |
| **Value** | $-30.9215$ | $1.6543$ | $0.3915$ | $0.0211$ | $0.0911$ | $-151.9032$ | $270.4504$ | $14.9551$ |

The corresponding results of one-step ahead prediction and 20-steps-ahead prediction are shown in Figures 8–11.

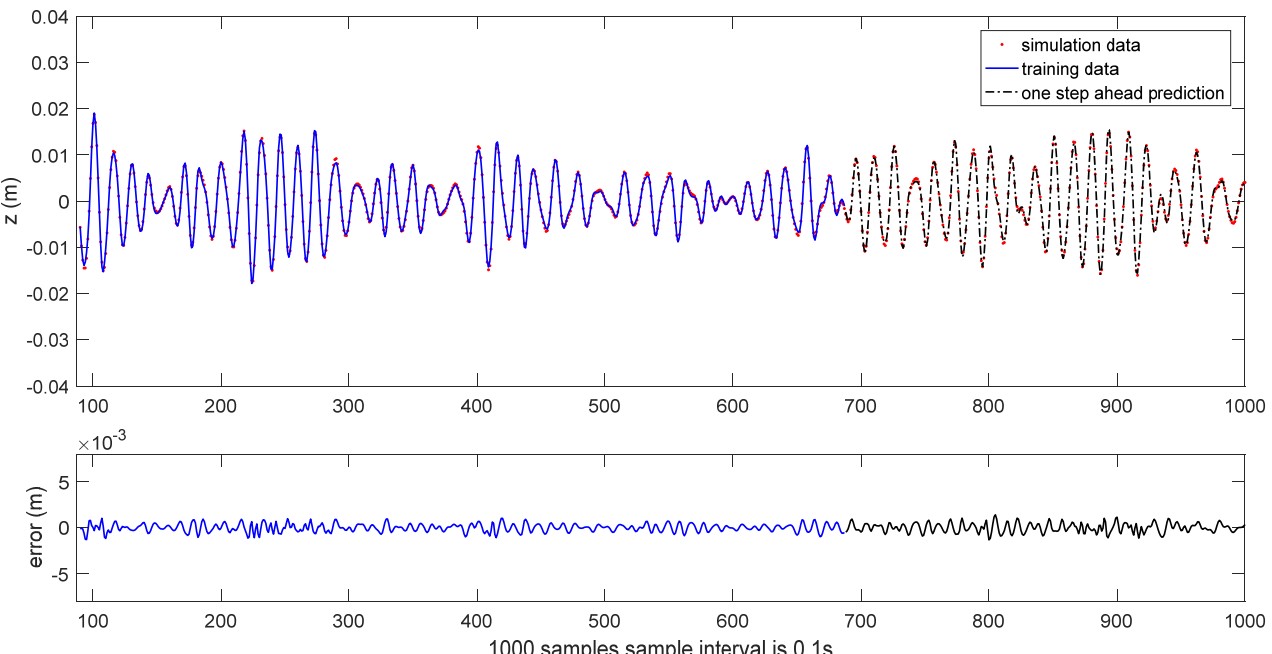

**Figure 8.** The one-step-ahead prediction of heave motion in irregular waves at the instant $k$ = 687 for the R-Class Icebreaker model.

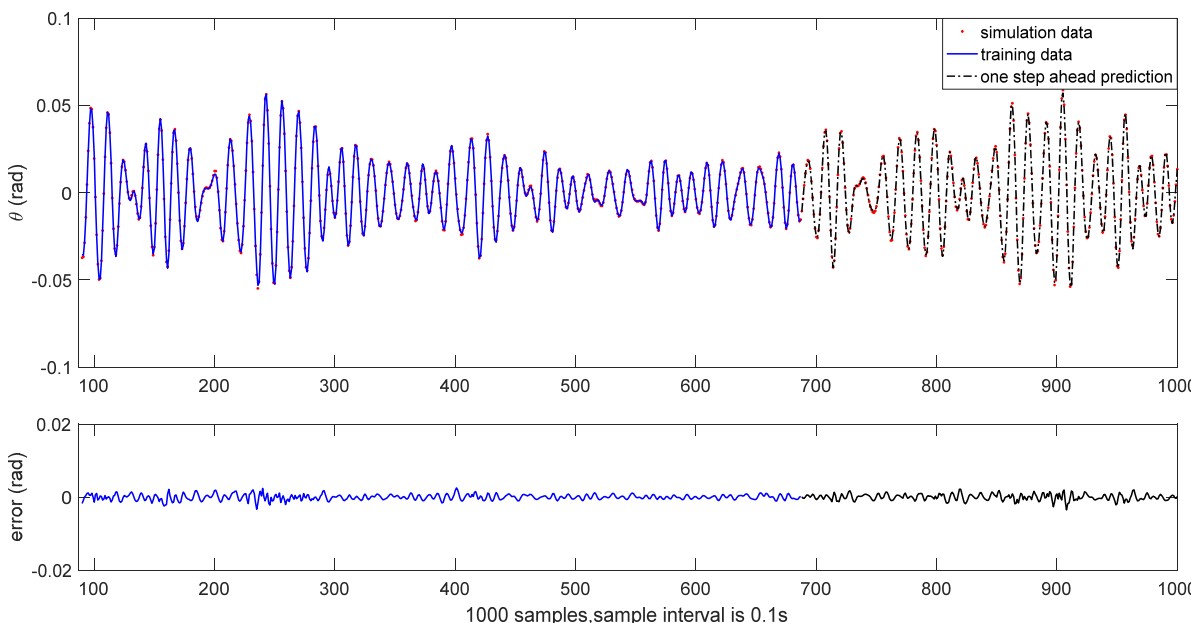

**Figure 9.** The one-step-ahead prediction of pitch motion in irregular waves at the instant $k$ = 687 for the R-Class Icebreaker model.

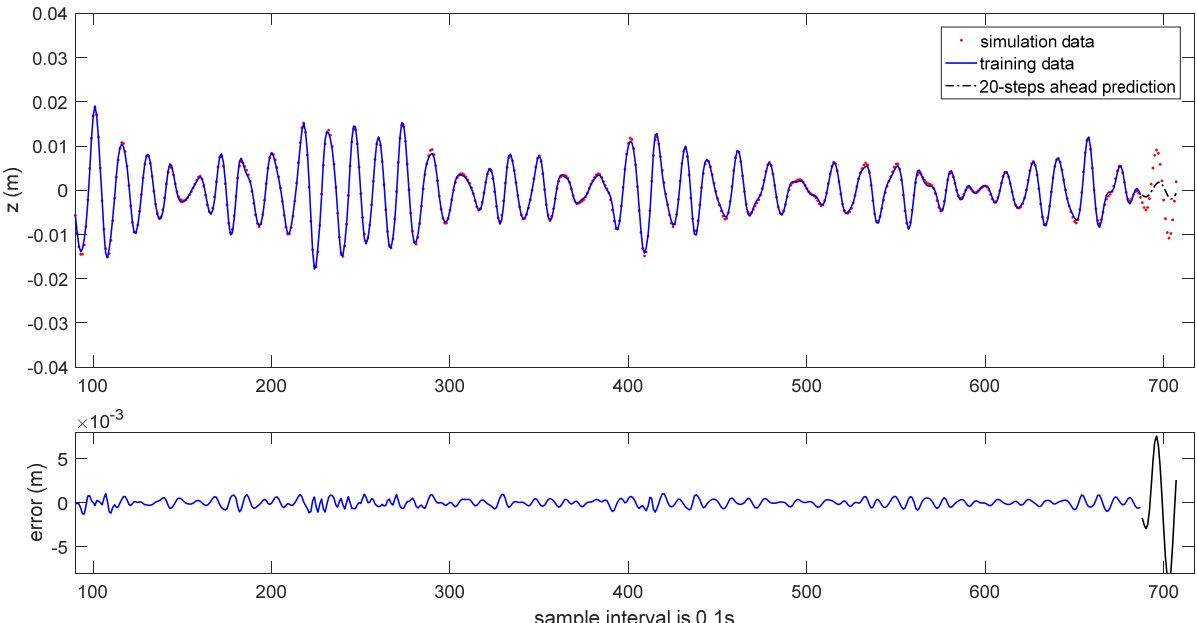

**Figure 10.** The 20-steps-ahead prediction of heave motion in irregular waves at the instant $k$ = 687 for the R-Class Icebreaker model.

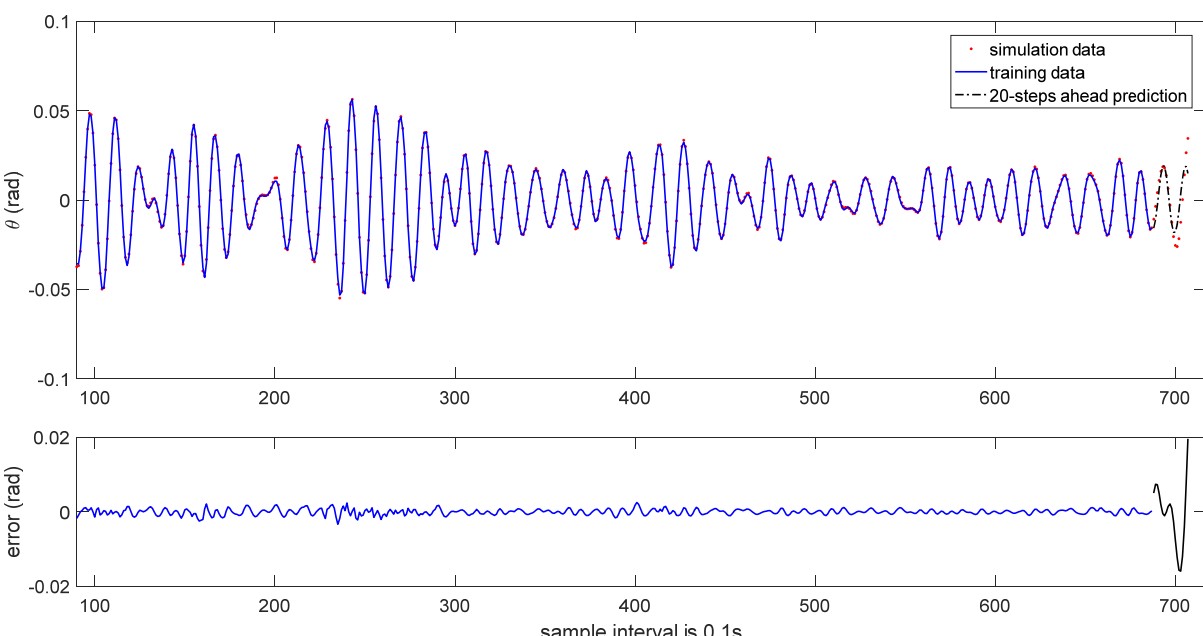

**Figure 11.** The 20-steps-ahead prediction of pitch motion in irregular waves at the instant *k* = 687 for the R-Class Icebreaker model.

The proposed method is an online modeling method. The structure and parameters of the CFT-FGWN can change online as the sliding data window moves. When the sliding data window receives new sample data, the wavelet model can be adjusted online over time. The dynamic feature of the modeling process is shown in Figure 12. It shows that several significant wavelet terms can characterize the coupled heave-pitch motions in irregular waves very well. It is also shown clearly that the number of significant wavelet terms changes after the coarse-tuning process, and it is different for heave motion and pitch motion. The number of the continuous fine-tuning does not exceed 5, and the continuous fine-tuning can also satisfy related modeling conditions, which shows that this modeling method is effective.

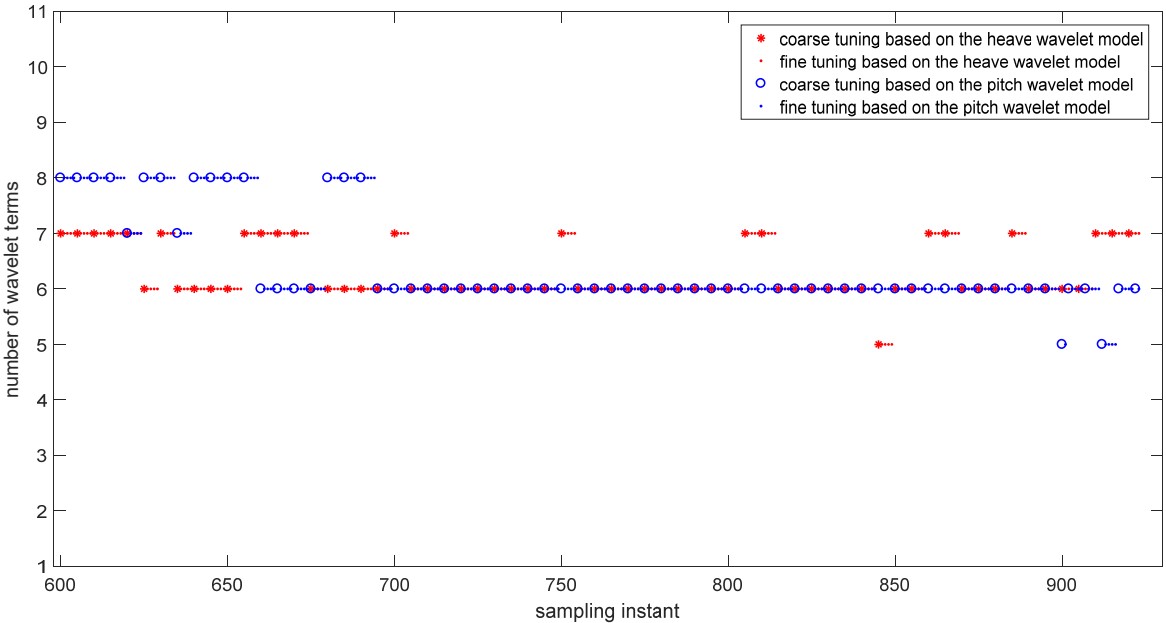

**Figure 12.** The structure of the CFT-FGWN changing over time based on the simulation data.

The computational time of the related modeling algorithm is shown in Table 7. Obviously, the coarse-tuning process needs more computational time than the fine-tuning process. Hence, the computational efficiency is improved by adding the continuous fine-tuning process.

**Table 7.** Computational time with simulation data.

|  | Number of Coarse Tuning | Number of Fine Tuning | Computational Time of Coarse Tuning | Computational Time of Fine Tuning |
|---|---|---|---|---|
| Heave-FGWN | 65 | 259 | 117.1726 s | 1.3241 s |
| Pitch-FGWN | 66 | 258 | 126.4073 s | 1.2001 s |

It is clearly shown that the CFT-FGWN has the ability to represent the coupled heave-pitch motions in irregular waves. As a NAR modeling method, it is not necessary to measure the information of waves. This is an especially beneficial advantage, since the state of the waves is usually difficult to measure online at the open sea. As long as the past state of the coupled heave-pitch motions is obtained, the forecast model can be established online. Besides, it is easy to see that different system inputs contribute differently to the system output; thus, the influences of different system inputs can be distinguished. This is another advantage of the proposed modeling method over the conventional neural network.

*4.2. Prediction Results Based on Experimental Data*

The experimental data of coupled heave-pitch motions of an FPSO model is also used to verify the effectiveness of the proposed modeling method. The experimental data are obtained from the State Key Laboratory of Ocean Engineering at Shanghai Jiao Tong University, and the principal particulars of the FPSO model are given in Table 8 [31]. The JONSWAP wave spectrum is used to actuate the FPSO model. The significant wave height and the peak spectral period are 0.1852 m and 1.68 s, respectively. There are 1000 samples, and the sampling interval is 0.08 s, as shown in Figure 13. The first three-fifths of the data are used for training, and the rest is used for testing. The coarsest resolutions $j_1$ and the finest resolutions $j_{\max 1}$ are set to 0 and 3 in the one-dimensional radial wavelets, and the coarsest resolutions $j_2$ and the finest resolutions $j_{\max 2}$ are set to 0 and 2 in the two-dimensional radial wavelets. The maximum number of the continuous fine-tuning is not larger than 5. The coarse training condition for the wavelet model of heave motion is that the other significant terms must be added in the model when $\sum_i ERR_i < 0.99$ or $RMSE_{training} > 0.008$, while the coarse training condition for the wavelet model of pitch motion is $\sum_i ERR_i < 0.99$ or $RMSE_{training} > 0.006$. These values are determined by the trial-and-error approach.

**Table 8.** Principal particulars of the FPSO model.

| Parameter | Value |
|---|---|
| Length between perpendiculars | 3.71 m |
| Breadth | 0.67 m |
| Depth | 0.32 m |
| Mean draft | 0.15 m |
| Displacement volume | 0.363 m$^3$ |

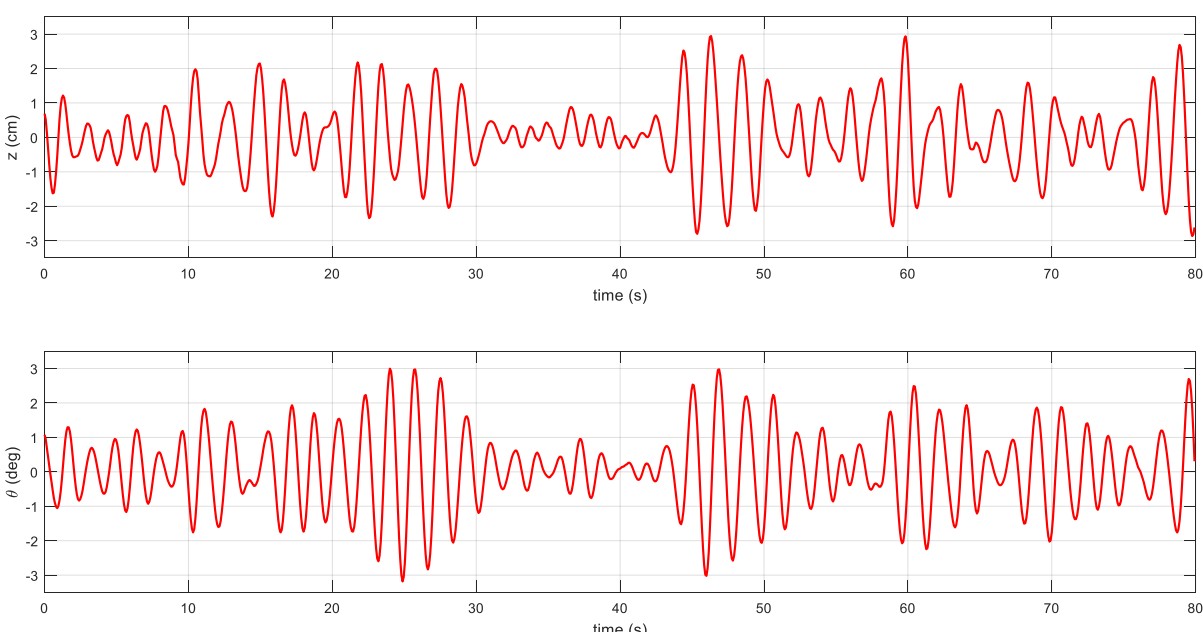

**Figure 13.** The coupled heave-pitch motions of the FPSO model in irregular waves.

The identified model is as follows:

$$
\begin{cases}
z(k) = c_1 \cdot \psi_{0;-1,4}^{[2]}(z(k-2), z(k-1)) + c_2 \cdot \psi_{2;3,-2}^{[2]}(z(k-2), z(k-1)) \\
\quad + c_3 \cdot \psi_{2;1}(z(k-2)) + c_4 \cdot \psi_{0;1,-1}^{[2]}(z(k-2), z(k-1)) \\
\quad + c_5 \cdot \psi_{2;7,0}^{[2]}(z(k-2), z(k-1)) + c_6 \cdot \psi_{2;-3,-3}^{[2]}(z(k-2), z(k-1)) \\
\quad + c_7 \cdot \psi_{3;10}(z(k-2)) \\
\theta(k) = d_1 \cdot \psi_{0;-1,4}^{[2]}(\theta(k-2), \theta(k-1)) + d_2 \cdot \psi_{2;3,-2}^{[2]}(\theta(k-2), \theta(k-1)) \\
\quad + d_3 \cdot \psi_{2;1}(\theta(k-2)) + d_4 \cdot \psi_{0;1,-1}^{[2]}(\theta(k-2), \theta(k-1)) \\
\quad + d_5 \cdot \psi_{2;6,0}^{[2]}(\theta(k-2), \theta(k-1)) + d_6 \cdot \psi_{3;-2}(\theta(k-1)) \\
\quad + d_7 \cdot \psi_{2;5}(\theta(k-2))
\end{cases}
\tag{17}
$$

where the related coefficients are given in Table 9. The outputs of the identified model are inversely normalized to obtain the original system outputs.

**Table 9.** The coefficients in Equation (17).

| Coefficient | $c_1$ | $c_2$ | $c_3$ | $c_4$ | $c_5$ | $c_6$ | $c_7$ |
|---|---|---|---|---|---|---|---|
| **Value** | $-37.5314$ | $0.4640$ | $0.0214$ | $-1.2207$ | $50.9184$ | $9.0101$ | $-0.0330$ |
| **Coefficient** | $d_1$ | $d_2$ | $d_3$ | $d_4$ | $d_5$ | $d_6$ | $d_7$ |
| **Value** | $-39.9604$ | $0.3860$ | $0.0167$ | $-1.1384$ | $1.8944$ | $0.0368$ | $0.0239$ |

The results of initial one-step ahead prediction are depicted in Figures 14 and 15. The results of 30-steps-ahead prediction are depicted in Figures 16 and 17. It can be seen that the prediction performance is satisfactory.

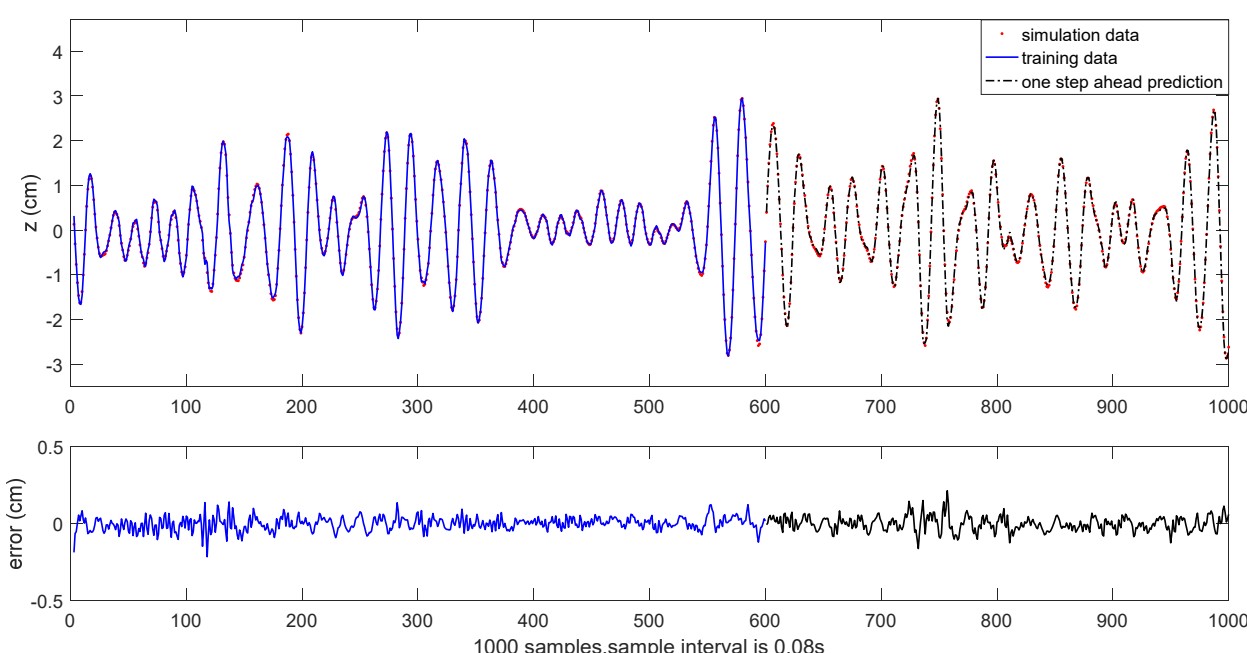

**Figure 14.** The one-step ahead prediction of heave motion in irregular waves for the FPSO model.

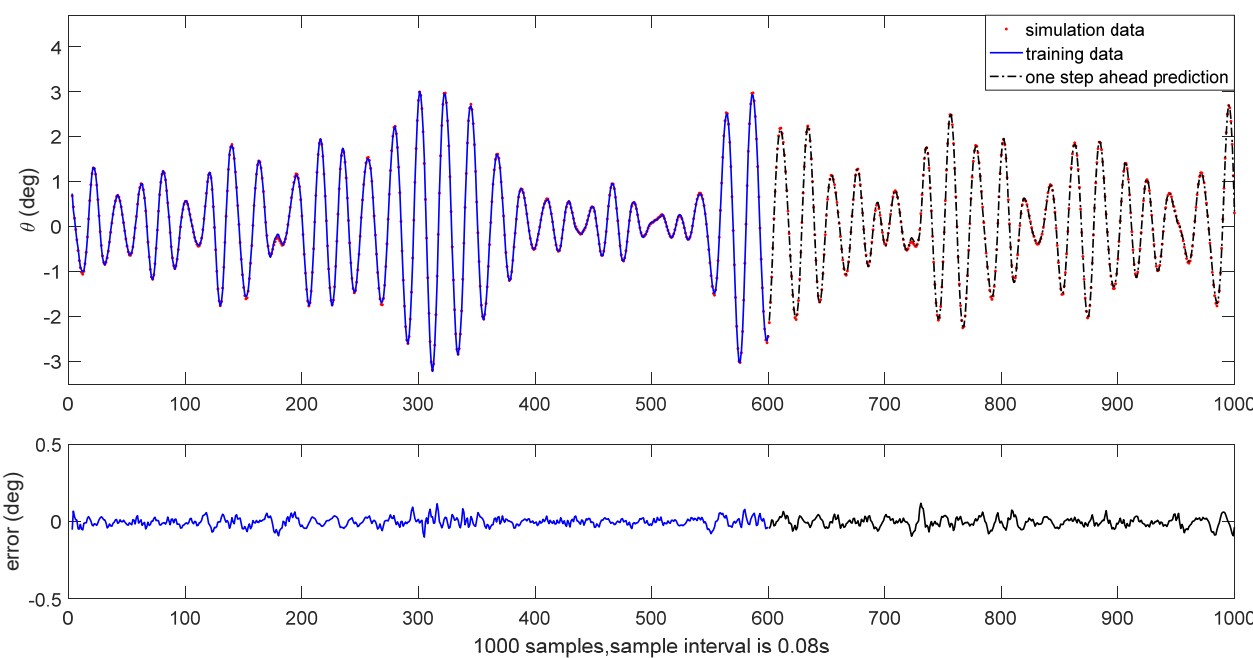

**Figure 15.** The one-step ahead prediction of pitch motion in irregular waves for the FPSO model.

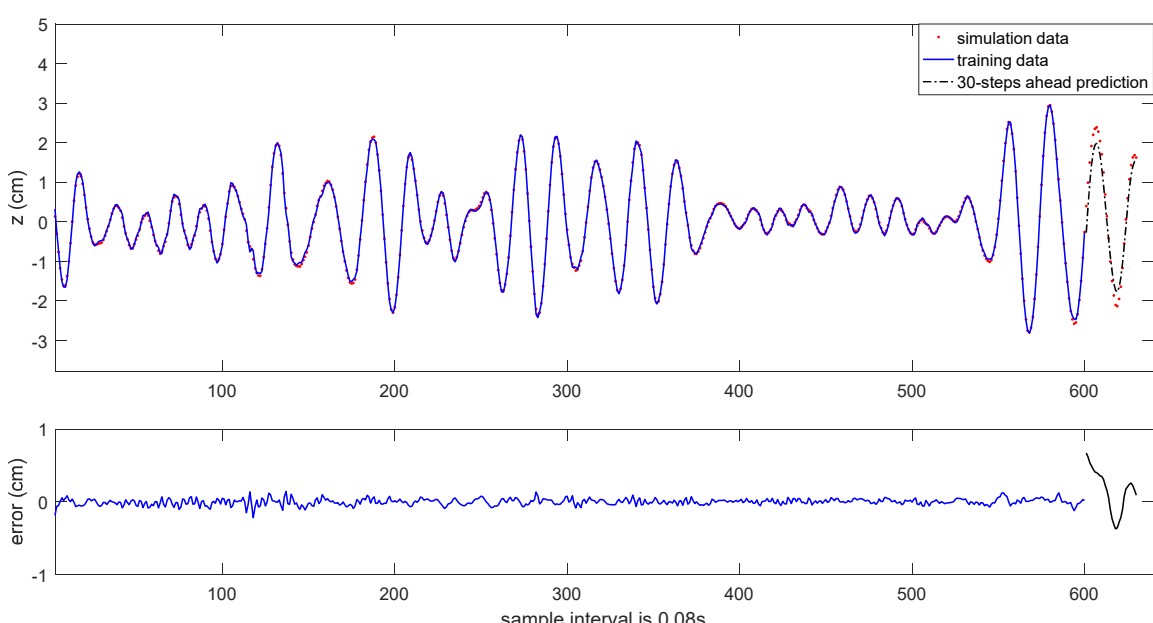

**Figure 16.** The 30-steps-ahead prediction of heave motion in irregular waves for the FPSO model.

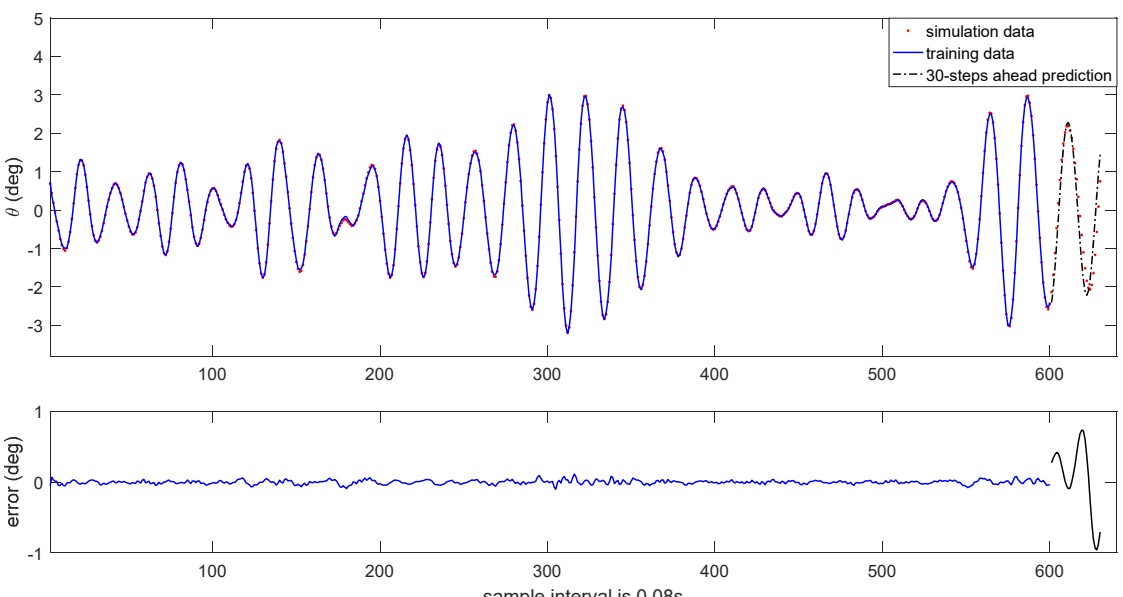

**Figure 17.** The 30-steps-ahead prediction of pitch motion in irregular waves for the FPSO model.

As the sliding data window continues to move, it receives new data and the model can be adjusted online. For example, when the sliding data window receives the data at the sampling instant $k = 763$, the corresponding model is identified as follows:

$$
\begin{cases}
z(k) = c_1 \cdot \psi_{0;-1,4}^{[2]}(z(k-2), z(k-1)) + c_2 \cdot \psi_{2;3,-2}^{[2]}(z(k-2), z(k-1)) \\
\qquad + c_3 \cdot \psi_{2;1}(z(k-2)) \ + c_4 \cdot \psi_{0;1,-1}^{[2]}(z(k-2), z(k-1)) \\
\qquad + c_5 \cdot \psi_{2;7,0}^{[2]}(z(k-2), z(k-1)) \ + c_6 \cdot \psi_{1;2,-3}^{[2]}(z(k-2), z(k-1)) \\
\qquad\qquad + c_7 \cdot \psi_{1;3,3}^{[2]}(z(k-2), z(k-1)) \\
\theta(k) = d_1 \cdot \psi_{0;-1,4}^{[2]}(\theta(k-2), \theta(k-1)) + d_2 \cdot \psi_{2;3,-2}^{[2]}(\theta(k-2), \theta(k-1)) \\
\qquad + d_3 \cdot \psi_{2;1}(\theta(k-2)) \ + d_4 \cdot \psi_{0;1,-1}^{[2]}(\theta(k-2), \theta(k-1)) \\
\qquad + d_5 \cdot \psi_{2;6,0}^{[2]}(\theta(k-2), \theta(k-1)) \ + d_6 \cdot \psi_{3;-2}^{[2]}(\theta(k-1)) \\
\qquad\qquad + d_7 \cdot \psi_{0;4,-2}^{[2]}(z(k-2), z(k-1))
\end{cases}
\tag{18}
$$

where the related coefficients are given in Table 10.

**Table 10.** The coefficients in Equation (18).

| Coefficient | $c_1$ | $c_2$ | $c_3$ | $c_4$ | $c_5$ | $c_6$ | $c_7$ |
|:---:|:---:|:---:|:---:|:---:|:---:|:---:|:---:|
| **Value** | $-36.5040$ | $-0.1151$ | $0.0192$ | $-1.4192$ | $42.8903$ | $1.9687$ | $0.0656$ |
| **Coefficient** | $d_1$ | $d_2$ | $d_3$ | $d_4$ | $d_5$ | $d_6$ | $d_7$ |
| **Value** | $-39.2299$ | $0.4820$ | $0.0175$ | $-1.0484$ | $2.2043$ | $0.0422$ | $-10.6728$ |

The results of the one-step ahead prediction and the 30-steps-ahead prediction are shown in Figures 18–21.

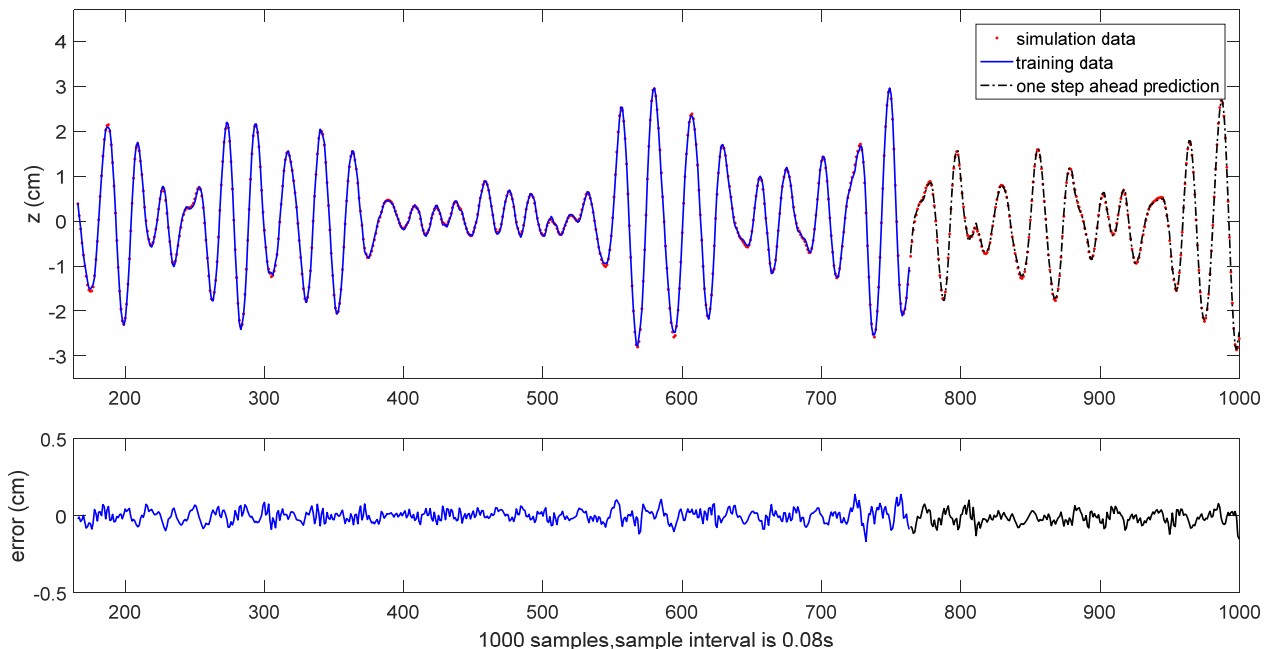

**Figure 18.** The one-step ahead prediction of heave motion in irregular waves at the instant $k = 763$ for the FPSO model.

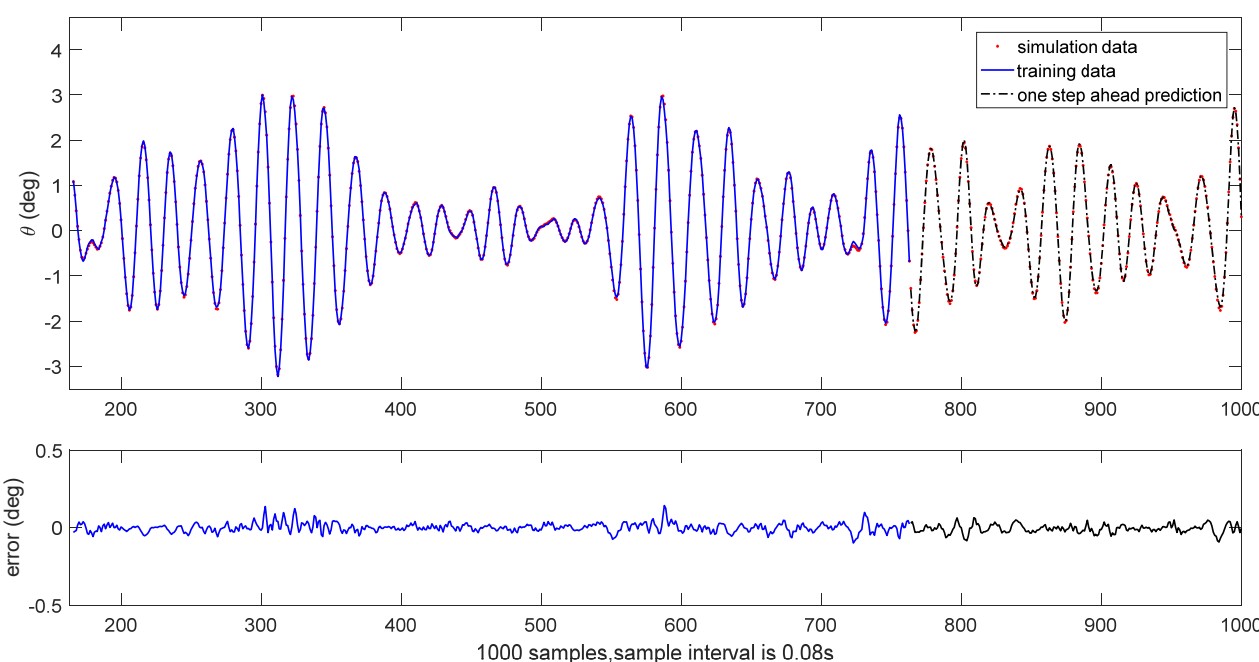

**Figure 19.** The one-step ahead prediction of pitch motion in irregular waves at the instant *k* = 763 for the FPSO model.

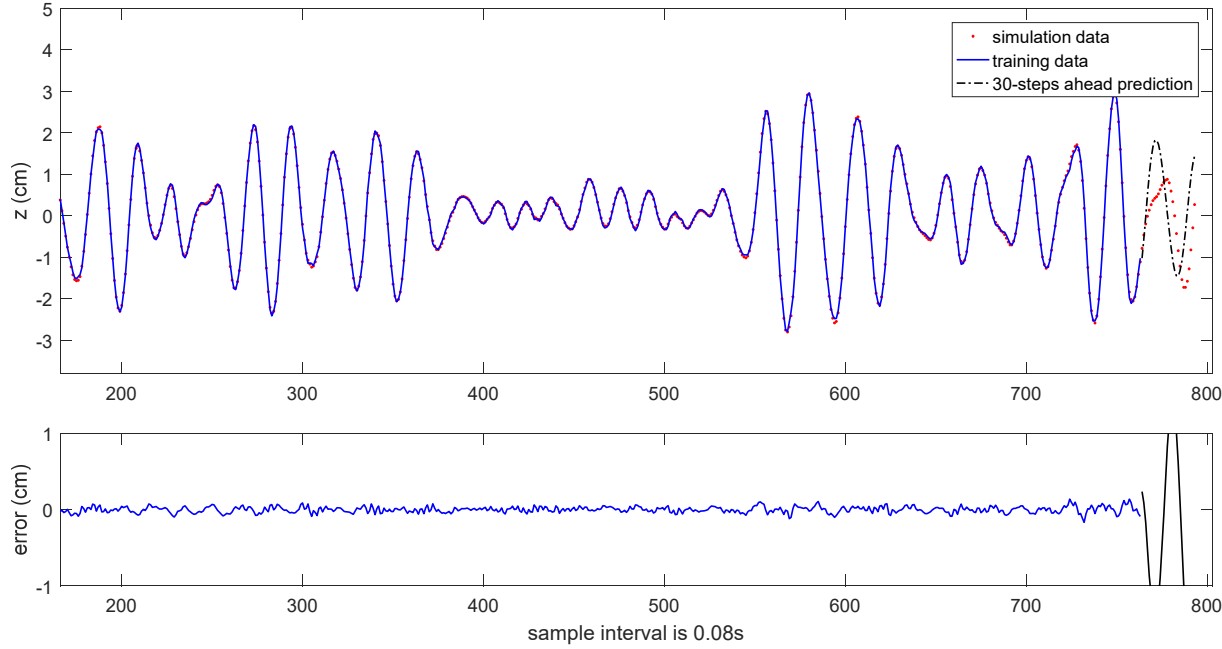

**Figure 20.** The 30-steps-ahead prediction of heave motion in irregular waves at the instant *k* = 763 for the FPSO model.

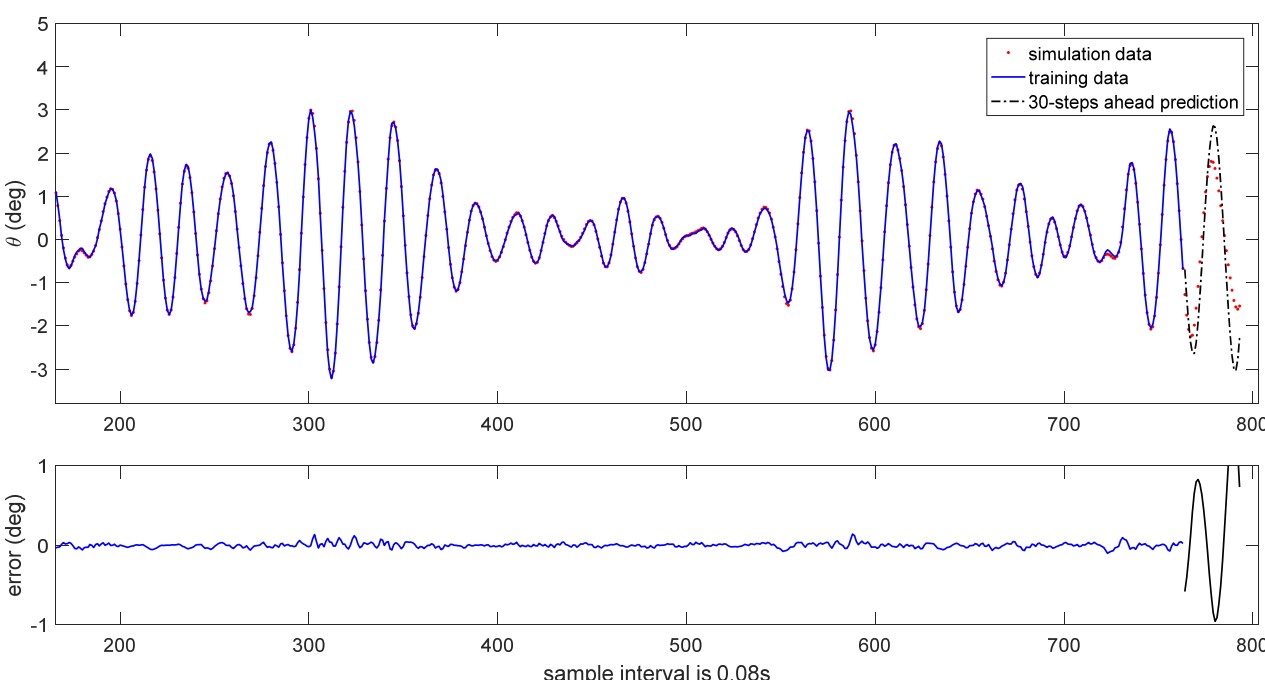

**Figure 21.** The 30-steps-ahead prediction of pitch motion in irregular waves at the instant *k* = 763 for the FPSO model.

The structure of the CFT-FGWN can be adjusted online over time, and the change process is shown in Figure 22. It can be seen clearly that several significant wavelet terms can characterize the coupled heave-pitch motions in irregular waves, which shows that the CFT-FGWN has a strong nonlinear fitting ability.

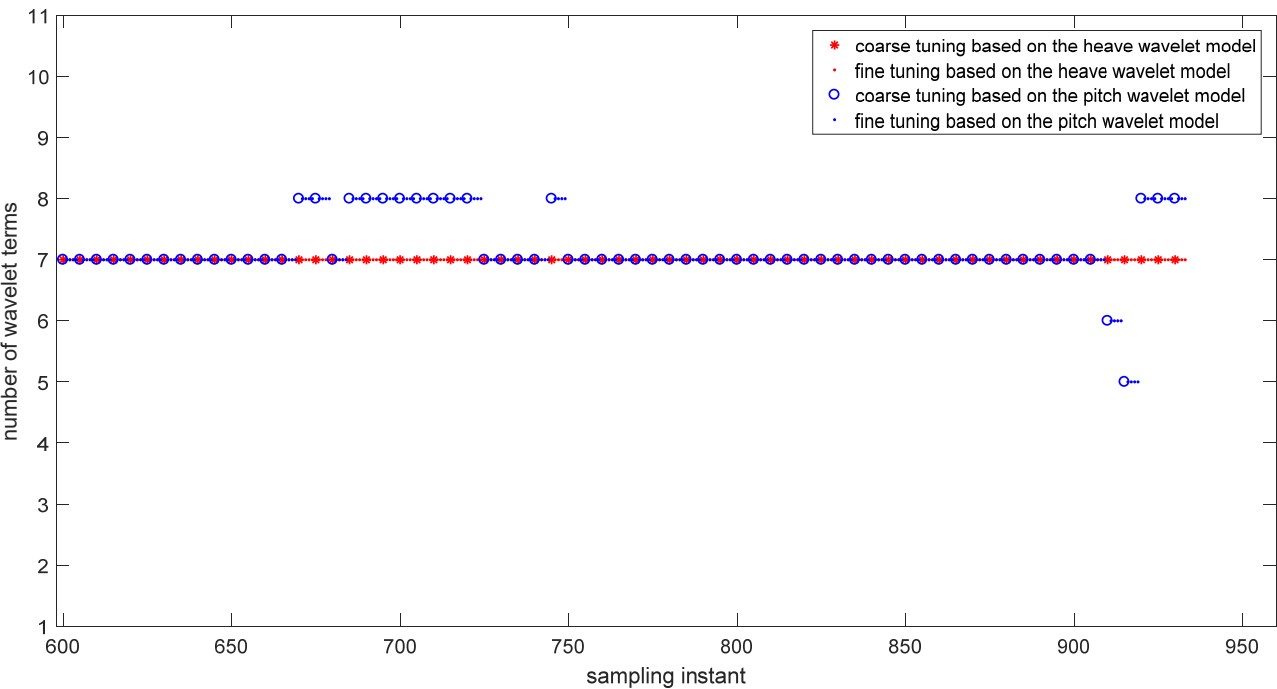

**Figure 22.** The structure of the CFT-FGWN changing over time based on the experimental data.

The computational time of the related modeling algorithm is given in Table 11. It can be seen that the fine-tuning process takes less computational time than the coarse-tuning process. The computational efficiency is improved significantly due to the continuous fine-tuning process.

**Table 11.** Computational time with experimental data.

|  | Number of Coarse Tuning | Number of Fine Tuning | Computational Time of Coarse Tuning | Computational Time of Fine Tuning |
|---|---|---|---|---|
| Heave-FGWN | 67 | 267 | 171.4563 s | 1.4676 s |
| Pitch-FGWN | 67 | 267 | 182.2069 s | 1.3377 s |

## 5. Conclusions

The CFT-FGWN is used for online modeling ship's coupled heave-pitch motions in irregular waves. The effectiveness of the modeling method is verified by applying it based on the simulation data and the real experimental data. The prediction results using the established wavelet model showed that this online modeling method is not only feasible, but also has high computational efficiency, and several significant wavelet terms can capture the intrinsic nonlinear dynamics of the coupled heave-pitch motions, which indicates that the modeling method has the ability to give a good prediction of the coupled heave-pitch motions of a ship in irregular waves, and it can be applied to MIMO systems.

Based on the simulation results, the continuous fine-tuning process requires less computational time, which increases the computational efficiency largely. A heuristic method will be considered to determine the number of continuous fine-tuning in the future work.

**Author Contributions:** Methodology and formal analysis, B.H. and Z.Z.; investigation, B.H.; writing-original draft preparation, B.H.; writing-review and editing, J.J. and Z.Z. All authors have read and agreed to the published version of the manuscript.

**Funding:** This work is financially supported by the National Natural Science Foundation of China (Grant Nos: 51509193, 51779140).

**Institutional Review Board Statement:** Not applicable.

**Informed Consent Statement:** Not applicable.

**Data Availability Statement:** Te data used to support the findings of this study are available from the corresponding author upon request.

**Conflicts of Interest:** The authors declare no conflict of interest.

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
