# Peer review of "Online Prediction of Ship Coupled Heave-Pitch Motions in Irregular Waves Based on a Coarse-and-Fine Tuning Fixed-Grid Wavelet Network"

_jmse, doi:10.3390/jmse9090989_

Round 1
Reviewer 1 Report
The problem with auto-plagiarism and lack of novelty, which was described in the previous review, has not been fully solved in the revised version. Please take into account that the concept, methods, definitions, and tools are the same as given in the previous papers. Using a continuous fine-tuning process and applying the previously published methods for a multi-dimensional system is interesting in comparison to the authors' previous works, but seems to be not enough for a new research paper. At least, please let the potential reader what exactly is new in this paper in comparison to the previous publications.
Other remarks have been already taken into consideration and necessary modifications are applied and thank you for that.
Author Response
Please find the attachment below.

Reviewer 2 Report
I still had difficulty to grasp the context of this work. The authors had been asked to explain better the utility of their work but such explanations were not found in their revised manuscript. The heave-pitch motions are the easiest to predict ship motions. What is the true benefit of the author's aproach has not been clear.
Author Response
Please find the attachment below.

Reviewer 3 Report
The authors have revised their papers and I do not have any other comments.
Author Response
Please find attachment below.

Round 2
Reviewer 2 Report
The context of the work is now described to some extent.